# Effect of simulated handball match-induced fatigue on isokinetic hamstring-to-quadriceps ratio and evertor-to-invertor ratio in professional players

Ali Shirani Dastjerdi[1], Hamed Esmaeili[1]*, Morteza Sadeghi[1], Behzad Bashiri[2]

1 Department of Sport Injuries and Corrective Exercises, Faculty of Sport Sciences, University of Isfahan, Isfahan, Iran, 2 Department of Biomedical Engineering, Faculty of Engineering, International University of Science and Technology in Kuwait, Ardiya, Kuwait.

* H.esmaeili@spr.ui.ac.ir

## Abstract

Knee and ankle injuries are the most common injuries in handball players. Many of these injuries occur toward the end of the game when fatigue sets in. The underlying mechanisms of this phenomenon are not well understood. This study investigated the effect of a handball match-simulation protocol on the hamstring-to-quadriceps (H:Q) and evertor-to-invertor (E:I) peak torque ratios. Thirty professional male handball players from Iranian super league teams participated in this study. Isokinetic concentric peak torque of the hamstrings, quadriceps, ankle invertor, and ankle evertors were measured at angular velocities of 60, 120, and 180 deg/s before and after the simulated handball match-induced fatigue protocol using a Biodex Isokinetic Dynamometer. Peak torque ratio of H:Q and E:I were calculated and compared between pre- and post-fatigue conditions. Fatigue significantly reduced H:Q peak torque ratio at 60 (p = 0.049), 120 (p = 0.002) and 180 (p = 0.014) deg/s, as well as E:I peak torque ratio at 60 (p = 0.010), 120 (p = 0.003) and 180 (p = 0.003) deg/s. These changes could contribute to an increased risk of anterior cruciate ligament (ACL) tears and lateral ankle sprains. Given the greater reduction in the H:Q peak torque ratio in hamstring muscles, targeted strengthening and improving the endurance of these muscles is recommended for professional handball players to mitigate fatigue effects.

## Introduction

Handball, as a complex and highly demanding intermittent sport, involves multiple high-intensity runs, frequent body contact, and other high-intensity actions to overcome opponents [1]. A handball match involves a large number of repeated accelerations, sprints, jumps, blocking, pushing and rapid changes in moving directions [2]. Sudden changes of direction, decelerations and accelerations, vertical and horizontal

**Competing interests:** The authors have declared that no competing interests exist.

jumps, throws, thrusts, pulls, and powerful, fast-paced clashes are among the movements of handball sport that have made it a sport with high risk of injury [3].

These physical demands are expected to lead to fatigue, which could be at least partially responsible for the reduction in the quantity and quality of high-intensity actions toward the end of a match compared to the beginning [4]. Fatigue can be categorized into two types: central fatigue, which originates within the central nervous system (CNS) and results in a reduced neural drive to the muscles, and peripheral fatigue, which involves processes at or distal to the neuromuscular junction that impair force generation [5–7]. Both forms of fatigue can appear during training sessions or matches, potentially leading to impaired coordination, muscle mechanical properties and neuromuscular activity, and a higher frequency of mistakes [8]. During elite male handball matches, there was a decrease in high-intensity running time, demanding actions, and time spent with a heart rate above 80% of maximum from the first half to the second [9]. Metabolic markers such as plasma free fatty acids, glycerol, glucose, and uric acid increased during the first half, with further increases in free fatty acids and glycerol during the second half [9]. These changes led to decline in sprint performance and reduction in vertical jump height compared to baseline levels, indicating significant fatigue development [9]. Moreover, handball match-induced fatigue has been shown to increase center of pressure area during landing and decrease maximum propulsion force for countermovement jump [10].

It has been shown that players who play 70 percent of the total match time experience changes such as reduction in high-intensity running, reduction in heart rate, reduction in technical and impaired physical performance, resulting from fatigue [8]. This condition could increase demand on musculoskeletal system and leaves players at high risk of injury. Notably, the highest number of injuries during six periods of international competitions occurred between minutes 16–30 of the first half and 46–60 of the second half [11].

Given these findings, it appears that physical fatigue deteriorates neuromuscular and sensorimotor control, which may elevate the risk of injury. Handball players report a higher prevalence of lower limb injuries than injuries to the upper limbs, neck, or head [11,12]. For instance, approximately 30% of injuries occur in the knee, and 25% in the ankle [12].

The hamstring-to-quadriceps (H:Q) peak torque ratio in the knee joint and evertor to invertor (E:I) peak torque ratio in the ankle are two risk factors used to assess muscle balance or imbalance in the knee and ankle [13–16]. Deterioration in these ratios may increase the risk of injury. A decreased H:Q ratio implies that the quadriceps are disproportionately stronger than the hamstrings, which may stress the anterior cruciate ligament (ACL) and may lead to injuries [17]. A decrease in the E:I peak torque ratio is linked to a higher risk of lateral ankle sprains [18].

Isokinetic muscle peak torque is a critical metric in assessing injury risk, particularly in sports and rehabilitation settings. Isokinetic testing can identify neuromuscular deficits, such as reduced torque production or delayed muscle activation, which are often linked to injury risk [19,20]. For example, a decline in hamstring peak torque has been associated with an increased risk of hamstring strains in soccer players [20].

Given the high incidence of lower limb injuries among handball players, particularly when players are fatigued during matches, it is crucial to investigate the effects of match-induced fatigue on H:Q and E:I peak torque ratios in handball. The aim of this study was to evaluate the effects of simulated handball match-induced fatigue on the isokinetic hamstring-to-quadriceps and invertor-to-evertor ratios in professional handball players.

## Materials and methods

### Participants

Thirty professional male handball players from Iranian Super League teams (age: 27 ± 3.1 years; height: 186.9 ± 7.3 cm; mass: 86.8 ± 6.7 kg) participated in this study. A prior power analysis revealed that for a statistical power of 0.9 at an alpha level of 0.05 and an effect size of 0.41, a sample size of at least 26 participants is required [21]. The effect size was chosen from a previous study that had used a similar fatigue protocol [22]. Participant recruitment for this study began on August 15, 2023, and concluded on November 30, 2023. The inclusion criteria for participation required all subjects to have engaged in competitive handball for a minimum of 5 years and to participate in more than thee training sessions per week. During the preseason, the players completed 8–10 training sessions per week, while during the season, they attended 5–7 sessions per week. All participants were active team members and participated in every match. Volunteers who reported injuries in the lower limbs in the last 6 months were excluded. All participants provided written informed consent. This study was approved by the University of Isfahan Ethics Committee and conducted in accordance with the Declaration of Helsinki.

### Testing preparation

Initially, participants were briefed on the testing procedure. The participants began with an 8-minute warmup on a stationary bicycle with no resistance at moderate velocity (70–80 revolutions per minute) [23]. Next, they sat on the dynamometer chair (Biodex Multi-joint System 3, Biodex Medical Systems, USA) which was adjusted so they had free and full movement in knee and ankle joints. Participants were fixed with two horizontal and vertical belts, and the test limb was fixed to the arm of the device. The seat of the device was adjusted to an angle of 100 degrees. The arm of the device was adjusted to a tilt angle of 0 deg. The movement axis of the knee was aligned parallel to the movement axis of the device. For the ankle test, the participant's ankle was positioned on the footplate and the foot was secured using two Velcro straps.

For each test session, the range of motion of ankle and knee joints was determined first [24]. Concentric isokinetic peak torque for quadriceps, hamstring, ankle invertors, and ankle evertors was measured at three different angular velocities of 60, 120 and 180 deg/s by a Biodex Multi-joint System 3 dynamometer (Biodex Medical Systems, USA) on the landing leg in a random order. The landing leg was defined as the leg that players use to land after jumping [25,26]. Each angular velocity was measured with one set of six repetitions. In each test, participants were instructed to perform the movements with maximum effort. Verbal encouragement was provided during the test. Peak torques were assessed at baseline and immediately after the handball-specific fatigue protocol to examine its effects on the outcomes.

### Fatiguing protocol

A previously used fatigue protocol was utilized to simulate match-induced fatigue [10] which included special movements during a handball match, such as sprinting, jumping, landing, sidestep cutting, and lateral and backward shifting (Fig 1, caption for more details). The protocol was divided into gradually increasing rounds, requiring participants to complete each round as quickly as possible. The number of rounds corresponded to the number of laps in the circuit (e.g., round I: 1 lap; round II: 2 laps; and so on). For each round, time of effort, heart rate using Polar F4 Blue ICE, and the rate of perceived exertion (RPE) using the 15-point Borg scale with a a range of 6–20 were recorded [27]. In this Borg Scale, a

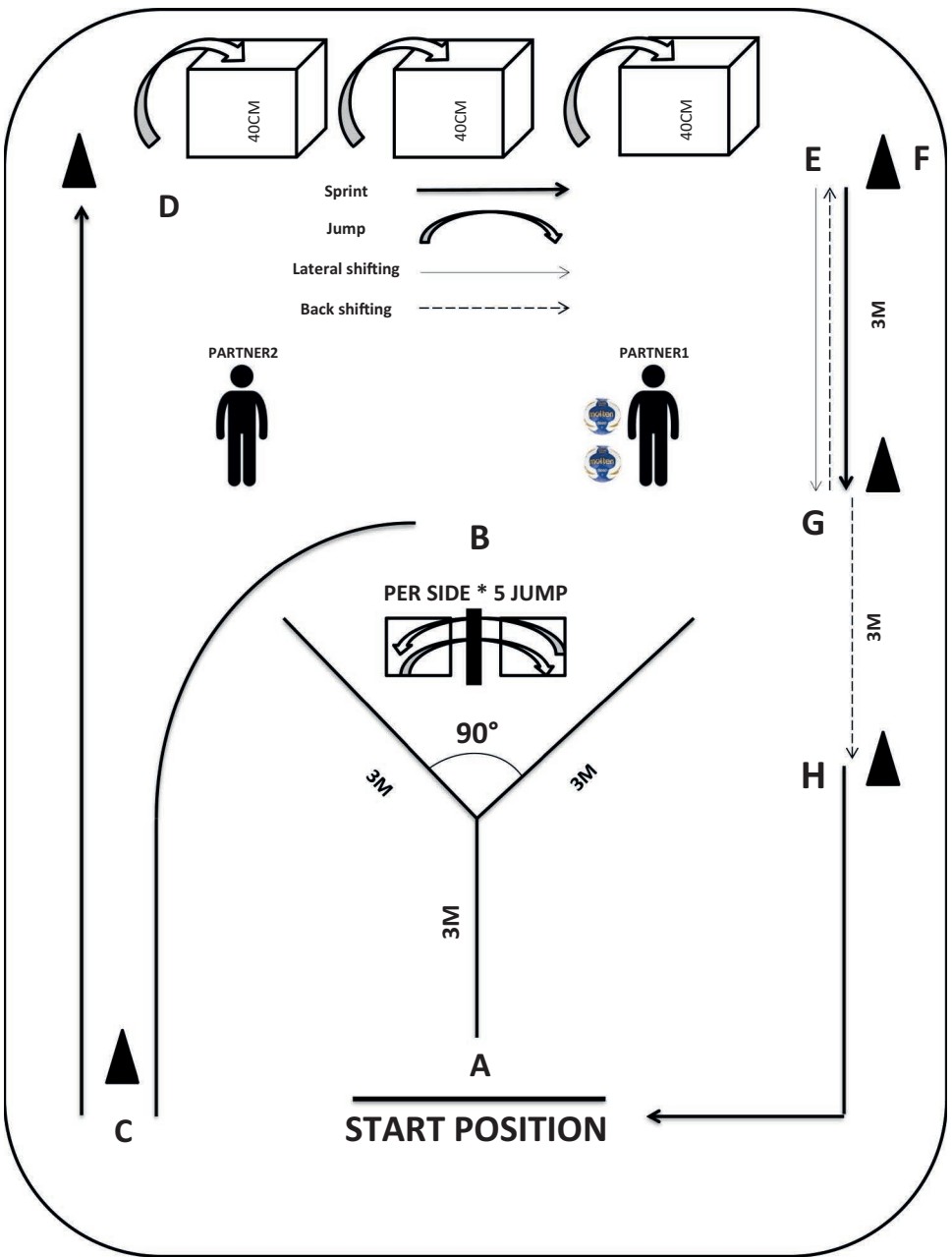

**Fig 1. Illustration of the fatigue protocol based on specific movements performed by handball athletes.** The athletes performed the following actions in all of the stations as fast as possible: **(A)** Acceleration and sidestep cutting maneuver for the right and run until the end of the 3-m distance; receive a pass (at the end of the 3-m length) from a partner positioned 2.5 m away (Partner 1); backshift until the 90°; run 3 m for the left side, and perform an accurate throw to a partner positioned 2.5 m away (Partner 2); this task was performed twice per lap; **(B)** Ten double-leg lateral jumping/ landing over a barrier (20-cm high), performed as fast as possible; **(C)** Sprint; **(D)** Double-leg plyometric jumps, performed as fast as possible; **(E)** Lateral shifting; **(F)** Sprint; **(G)** Back Shifting; and **(H)** Sprint with a 90° change of direction. The protocol was divided into rounds, where the round number represents the number of laps in the circuit (e.g., round I: 1 lap, round II: 2 laps, round III: 3 laps, and so on) until the participant is unable to continue performing the protocol.

score of six indicates no fatigue, while scores between 17 and 20 indicate the inability to continue the exercises. During the trials, if participants reported a score of 17 or higher at the end of a round, the fatigue protocol ended. If they reported

a lower score, they performed another round until they reached a score of 17. Borg scale has been shown to have high test-retest reliability [28,29].

## Data analysis

Peak knee flexion, knee extension, ankle inversion and ankle eversion torque were recorded for each repetition. Between each set of the test, participants rested for 2-min. To calculate the H:Q ratio, peak flexion torque was divided by the peak extensor torque [23]. To calculate the E:I ratio, peak eversion torque was divided by the peak inversion torque [18]. Peak torques per body mass, along with the peak torque ratios of H:Q and E:I, were used in subsequent analyses.

## Statistical analysis

Shapiro-Wilk test was used to ensure normality of data distribution. To investigate the effect of fatigue on the outcomes, a 2 (condition: pre- and post-fatigue) × 3 (angular velocity: 60, 120, and 180 deg/s) repeated measures ANOVA was used. Holm-Bonferroni adjustment was used for multiple comparisons. Effect size (ES), reported using partial eta squared, $\eta_p^2$, was classified as small ($\eta_p^2 = 0.01$), medium ($\eta_p^2 = 0.06$), and large ($\eta_p^2 = 0.14$) [30]. All statistical analyses were performed at a significance level of $p < 0.05$ using SPSS software, version 18.0 (SPSS Inc, Chicago IL, USA).

## Results

### Fatigue markers

Each participant executed a particular number of laps to achieve fatigue, ranging between four and eleven ($5.8 \pm 2.4$ laps or $276 \pm 79.2$ s). Significant increases were observed for heart rate ($F_{(1,29)} = 341.58$; $p < 0.001$) and RPE ($F_{(1,29)} = 392.42$; $p < 0.001$) (Table 1).

### Effect of fatigue on peak torques

The results showed that the main effect of fatigue ($F_{(1,29)} = 36.14$, $p < 0.01$, ES = 0.56) and the main effect of angular velocity ($F_{(2,58)} = 87.58$, $p < 0.01$, ES = 0.751) were significant, but the interaction effect was not significant ($F_{(2,58)} = 2.20$, $p = 0.120$, ES = 0.070) on the hamstring peak torque to body mass ratio. Multiple comparisons showed that fatigue decreased hamstring peak torque in all angular velocities (Table 2).

For the quadriceps peak torque to body mass ratio, the results showed significant main effects for fatigue ($F_{(1,29)} = 7.18$, $p = 0.012$, ES = 0.198) and angular velocity ($F_{(2,58)} = 296.68$, $p = 0.001$, ES = 0.911), and a significant fatigue × angular velocity interaction effect ($F_{(2,58)} = 4.20$, $p = 0.020$, ES = 0.127). This indicates that fatigue affects quadriceps peak torque differently at various angular velocities. Post-hoc tests revealed that fatigue was associated with a decrease in quadriceps peak torque at 60 deg/s and 180 deg/s, but not at 120 deg/s (Table 2).

Regarding the invertor peak torque to body mass ratio, the results of ANOVA showed a significant main effect of fatigue ($F_{(1,29)} = 12.97$, $p = 0.001$, ES = 0.309) and angular velocity ($F_{(2,58)} = 71.46$, $p = 0.001$, ES = 0.711), but no significant fatigue × angular velocity interaction ($F_{(2,58)} = 2.11$, $p = 0.130$, ES = 0.068). Multiple comparisons showed that, after fatigue, invertor peak torque decreased only in 60 deg/s (Table 3).

**Table 1. Mean and standard deviation (±) of the fatigue parameters.**

|  | Baseline | Fatigue State | F | P |
|---|---|---|---|---|
| **Heart rate** | 64.26 ± 8.2 | 186.31 ± 7.4 | 341.58 | **< 0.001** |
| **RPE** | 6 ± 0 | 20 ± 0 | 392.42 | **< 0.001** |

**Bold** values indicate significant difference in $p \le 0.05$

Repeated measures analysis of variance showed significant main effects of fatigue ($F_{(1,29)}$ = 44.01, p = 0.001, ES = 0.603) and angular velocity ($F_{(2,58)}$ = 99.48, p = 0.001, ES = 0.774), and a significant fatigue × angular velocity interaction ($F_{(2,58)}$ = 3.35, p = 0.042, ES = 0.104) on the evertor peak torque to body mass ratio. The handball-specific fatigue protocol was associated with decrease in evertor peak torque in all three angular velocities (Table 3).

### Effect of fatigue on peak torque ratios

ANOVA test showed significant main effects of fatigue ($F_{(1,29)}$ = 18.143, p = 0.001, ES = 0.385) and angular velocity ($F_{(2,58)}$ = 60.019, p = 0.001, ES = 0.674), but no significant fatigue × angular velocity interaction effect ($F_{(2,58)}$ = 0.526, p = 0.594, ES = 0.018) on H:Q peak torque ratio. Fig 2 illustrates H:Q peak torque ratio in the pre- and post-fatigue in different angular velocities. As shown, fatigue decreased H:Q ratio in 60 ($t_{(1,29)}$ = 2.662, p = 0.049), 120 ($t_{(1,29)}$ = 4.202, p = 0.002) and 180 ($t_{(1,29)}$ = 3.327, p = 0.014) deg/s.

ANOVA test showed significant main effects for fatigue ($F_{(1,29)}$ = 17.39, p = 0.001, ES = 0.375) and angular velocity ($F_{(2,58)}$ = 29.90, p = 0.001, ES = 0.508), but no significant fatigue × angular velocity interaction effect ($F_{(2,58)}$ = 1.63, p = 0.206, ES = 0.053) on E:I. As shown in Fig 3, fatigue decreased E:I in 60 ($t_{(1,29)}$ = 3.469, p = 0.010), 120 ($t_{(1,29)}$ = 4.164, p = 0.003) and 180 ($t_{(1,29)}$ = 4.198, p = 0.003) deg/s.

## Discussion

Knee and ankle injuries are the most common injury sites among handball players, especially when players experience fatigue during matches. The effect of match-induced fatigue on hamstring, quadriceps, ankle invertors and evertors, and their subsequent ratios is not well understood. The aim of the current study was to investigate the effect of simulated handball match-induced fatigue on H:Q and E:I peak torque ratios in Iranian handball Super League players. Our results

**Table 2. Comparison of hamstring and quadriceps peak torque ratio (in Nm/body mass) in the pre- and post-fatigue at different angular velocities. Data are presented in Mean ± SD.**

|  |  | Pre | Post | Mean Difference | Standard Error | 95% Confidence Interval | t | $p_{holm}$ |
|---|---|---|---|---|---|---|---|---|
| Hamstring | 60 deg/s | 1.83 ± .36 | 1.56 ± .25 | 0.27333 | 0.0649 | (0.476 - 1.330) | 4.214 | **0.001** |
|  | 120 deg/s | 1.55 ± .24 | 1.43 ± .26 | 0.12247 | 0.0360 | (0.280 - 1.077) | 3.398 | **0.006** |
|  | 180 deg/s | 1.32 ± .23 | 1.18 ± .23 | 0.13907 | 0.0198 | (0.823 - 1.808) | 7.031 | **0.001** |
| Quadriceps | 60 deg/s | 2.84 ± .35 | 2.65 ± .43 | 0.1827 | 0.0649 | (0.186 - 0.961) | 2.81 | **0.026** |
|  | 120 deg/s | 2.33 ± .34 | 2.24 ± .31 | 0.0852 | 0.0564 | (-0.092 - 0.638) | 1.51 | 0.141 |
|  | 180 deg/s | 1.68 ± .25 | 1.60 ± .27 | 0.9749 | 0.0668 | (0.036 - 0.772) | 2.19 | **0.041** |

**Bold** values indicate significant difference in p ≤ 0.05.

**Table 3. Comparison of invertor and evertor peak torque ratio (in Nm/body mass) in the pre- and post-fatigue at different angular velocities. Data are presented in Mean ± SD.**

|  |  | Pre | Post | Mean Difference | Standard Error | 95% Confidence Interval | t | $p_{holm}$ |
|---|---|---|---|---|---|---|---|---|
| Invertor | 60 deg/s | 0.44 ± .10 | 0.39 ± .10 | 0.0469 | .01233 | (0.290 - 1089) | 3.80 | **.003** |
|  | 120 deg/s | 0.37 ± .08 | 0.35 ± .09 | 0.0240 | .01194 | (-0.006 - 0.734) | 2.01 | .161 |
|  | 180 deg/s | 0.31 ± .09 | 0.30 ± .06 | 0.0165 | .01179 | (-0.111–0.617) | 1.40 | .345 |
| Evertor | 60 deg/s | 0.42 ± .12 | 0.33 ± .08 | 0.08779 | .01389 | (0.684–1.612) | 6.319 | **.001** |
|  | 120 deg/s | 0.35 ± .09 | 0.27 ± .06 | 0.07345 | .01292 | (0.586–1.478) | 5.685 | **.001** |
|  | 180 deg/s | 0.27 ± .08 | 0.21 ± .05 | 0.05667 | .01202 | (0.435–1.276) | 4.714 | **.001** |

**Bold** values indicate significant difference in p ≤ 0.05

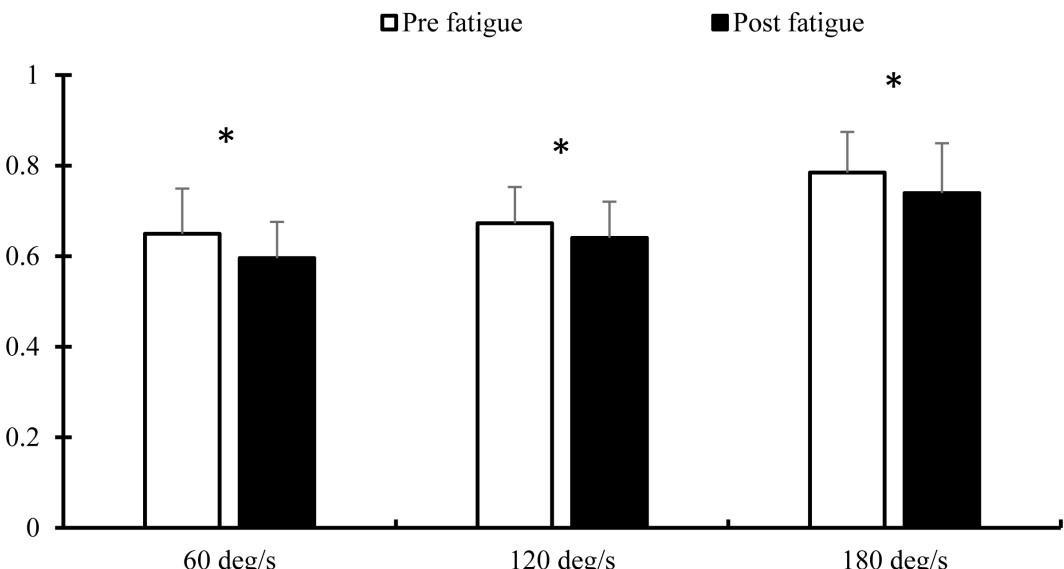

**Fig 2. Comparison of H:Q peak torque ratio at different angular velocities in pre- and post-fatigue (Mean and SD).** (*) stands for significant difference in p ≤ 0.05.

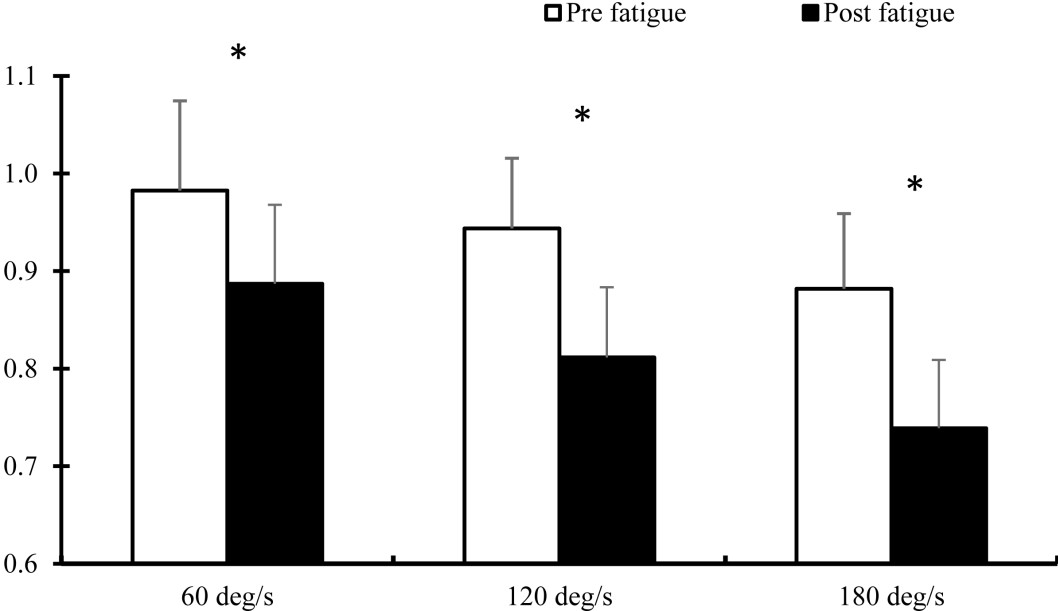

**Fig 3. Comparison of E:I peak torque ratio in different angular velocities in pre-and-post fatigue (Mean and SD).** (*) stands for significant difference in p ≤ 0.05.

showed that fatigue decreased peak hamstring and quadriceps muscles torque per body mass and the H:Q peak torque ratio. Also, the results showed that fatigue was associated with a decrease in peak invertor and evertor muscles torque per body mass and decrease in E:I peak torque ratio. In line with our results, Pinto *et al*. and Sangnier and Tourny-Chollet showed decreases in knee flexor and extensor torque and hamstring-to-quadriceps ratios [16,31]. Pinto et al

showed that after completing a 30-repetition isokinetic fatigue test protocol in professional soccer players, the knee extensor and flexor moments decreased and the H:Q was decreased at the end of the test due to a significant reduction in knee flexor moment as well [31]. Sangnier and Tourny-Chollet also using an isokinetic fatigue protocol on professional soccer players showed decreased strength of hamstring and quadriceps muscles [16]. Massamba *et al*. showed that after a 5-min all-out exercise training-induced fatigue, the maximum voluntary isometric force of quadriceps and hamstring muscles of healthy men decreased [32]. Our observed decreases in hamstring and quadriceps muscle peak torques in the current study, indicate effect of a simulated handball match-induced fatigue on knee flexor and extensor muscles strength and force output. While it has been well known that fatigue affects peak force output, training modalities seem to have different effects on force output reduction [33–37]. After completion of different fatigue protocols, similar reductions in force output are observed in knee flexor and extensor muscles concentric or isometric measurements [32,34]. The decrease in strength and torque output may result from a reduced ability of the muscles to generate force, which is associated with a decrease in the number of cross-bridges generating force and the number of cross-bridges actively working in the muscle [38]. This reduction can result from either general or localized muscle fatigue. Fatigue has both central and peripheral origins [6]. Peripheral fatigue is characterized by inability of the muscle to contract effectively [39]. Central fatigue, on the other hand, involves a significant reduction in the neural drive to the muscles [6], typically occuring after prolonged efforts. Therefore, it is plausible that the fatigue induced by our protocol has peripheral origins. However, according to the central governor model [40], the central component may also play a role in this type of exercise. This model suggests that the central nervous system interacts with the muscles to protect vital organs (e.g., the heart and brain) by reducing neural drive in high-demand situations, even during short-term exercises [41]. Thus, the fatigue protocol used in the present study may induce both peripheral and central fatigue.

The proposed fatigue protocol was designed to induce high levels of physiological and psychological stress, ultimately leading to exhaustion. This was demonstrated by the heart rate (HR) recorded at the point of exhaustion, which reached $186.31 \pm 7.4$ bpm, approximately 96% of the predicted maximum HR. Additionally, participants reported maximum values on the RPE score of 20, further confirming the extreme cardiologic stress induced by the protocol. These findings, along with the observed reductions in quadriceps, hamstring invertor and evertor muscles peak torque per body mass and their respective ratios, strongly suggest that the protocol effectively induces fatigue. These results indicate that our protocol effectively simulated fatigue in handball players, making it a valuable tool for both training and research purposes.

Our results showed that knee flexor muscles are influenced by handball match induced fatigue as well as knee extensors. Changes in fatigued condition are repetitively reported for knee extensor muscles [42,43]. However, our results are inconsistent with those of Marshal *et al.* who showed soccer simulation match-induced fatigue had no effect on hamstring muscles peak torque [42]. This inconsistency could be attributed to differences in the population and fatigue protocol. Marshal *et al.* studied soccer players and a soccer simulation match, whereas the current study included professional handball players and used a handball match-simulating protocol which engaged both hamstring and quadriceps muscles.

Our results showed that H:Q peak torque ratio decreased after fatigue protocol. The reduction in this ratio was associated with a decrease in hamstring and quadriceps muscles peak torques. Since this ratio is calculated by dividing hamstring peak torque by quadriceps peak torque, it seems that hamstring muscles are more affected by fatigue. Pinto *et al.* showed that after a fatigue protocol, hamstring muscles torque decreased to a greater extent than that of quadriceps [31], which is in line with the results of our study. Hamstring muscles besides their complex muscular architecture and multi-joint function, comprise a significant number of type II fibers and are less utilized in most daily activities, making them more susceptible to muscle damage and fatigue [44,45]. Although the mechanism of hamstring and ACL injuries is still a subject of debate, it is well-established that neuromuscular fatigue is an important risk factor [46]. For example, changes in hamstring muscle coordination, alterations in hamstring length during late swing phase of running cycle, and altered ACL-hamstring reflex are associated with muscle stiffness regulation and knee stability after a fatigue protocol [31]. Additionally, it has been shown that a decreased H:Q is associated with increased loading on the forefoot and toe regions [47]. The increased loading could

lead to increased risk of stress fracture in the forefoot region [48]. Overall, these findings also indicate that knee flexors should be well-trained to enhance their force production capacity and endurance, thus reducing susceptibility to hamstring and ACL injuries. Additionally, a reduction in the H:Q peak torque ratio increases the risk of ACL injury [16].

The results of the present study demonstrated that following a simulated handball game fatigue protocol, invertor and evertor muscles peak torque decreased. Additionally, a decrease in E:I muscles peak torque ratio was observed in the post-fatigue condition. The reduction in isokinetic maximum torque of the invertor and evertor muscles can be attributed to the effects of the fatigue protocol, as previously mentioned. These changes in muscle torques have implications for foot and ankle function. Lin and colleagues, using an isokinetic concentric-eccentric fatigue protocol, investigated the effects of invertor and evertor muscles fatigue and found that it led to a decrease in ankle joint position sense in the frontal plane [49]. Similarly, Rodriguez et al. demonstrated that after inducing fatigue by maintained eversion and plantar flexion at 70% of the maximum voluntary isometric contraction until a 10% decline in force, neuromuscular control was compromised in fatigued evertor muscles [50]. Bars Silva *et al.* induced fatigue through active localized ankle eversion exercises against resistance and demonstrated that exercise-induced muscle fatigue significantly increases the reaction time of the peroneal muscles [51]. This delayed reaction time can impair the muscles' ability to respond quickly to sudden changes in ankle position, potentially increasing the risk of ankle injuries [51]. Sandrey and Kent used a fatigue protocol targeting the ankle evertors, followed by tests to measure their ability to detect and replicate specific ankle positions in the frontal plane [52]. They also assessed absolute error in joint position sense, as an indicator of proprioceptive depth perception; they demonstrated that eversion fatigue negatively affected the ankle's ability to sense its position in the frontal plane [52]. Castillo et al. found that fatigue in the invertor and evertor muscles significantly impaired performance in functional jump tests [53]. They reported that participants experienced reduced jump height, distance, and control after muscle fatigue, indicating a decline in dynamic stability and power output [53]. Thus, fatigue-induced reductions in evertor muscle peak torque can decrease proprioceptive sensation and increase the risk of ankle sprains

Our results indicate that fatigue led to a decrease in the E:I peak torque ratio. This decrease in the E:I peak torque ratio is associated with a higher risk of lateral ankle sprains [18]. As shown, the decline in evertor muscle torque had greater contribution to decrease in E:I peak torque ratio than that of invertor muscles. It has been shown that prior to the occurrence of lateral ankle sprains, the E:I torque ratio decreases [18,54]. In our study, fatigue was associated with decreases in evertor muscles peak torque at 60, 120, and 180 deg/s. However, the effect of fatigue on invertor muscles was different. Specifically, fatigue only decreased invertor muscle peak torque at 60 deg/s, with no effect at 120 and 180 deg/s. Based on these results, it seems that fatigue induced by a handball simulation protocol could contribute to an increased risk of lateral ankle sprains. The E:I peak torque ratio is derived from dividing the peak torque of the evertor muscles by that of the invertor muscles. Given that the reduction in evertor muscle torque is greater than that of the invertor muscles, the decrease in this ratio is likely due to the greater decline in evertor muscle torque. Therefore, special attention should be considered to the endurance capacity and strength of the evertor muscles during sports exercises to minimize the impact of fatigue and reduce the risk of lateral ankle sprains, which are among the most common injuries in handball players [11].

The large effect size ($\eta_p^2 > 0.14$) demonstrates that the simulated handball game fatigue protocol had a robust and clinically significant impact on knee and ankle musculature peak torques. This large effect size suggests that the simulated handball game fatigue protocol induced significant changes in muscle performance, as measured by peak torques. This is clinically relevant because peak torque is a key indicator of muscle strength and function, which are critical for athletic performance and injury prevention in handball players. For coaches and sports scientists, the large effect size underscores the importance of incorporating fatigue protocols into training regimens to better prepare athletes for the physical demands of competition. By replicating game-like fatigue conditions, athletes can develop greater resilience and adaptability in their knee and ankle musculature.

Despite the promising findings of this study, some limitations should be acknowledged. First, we limited the measurements to concentric contrations to calculate conventional ratios. However, a previous study have shown that the behavior

of eccentric torque is similar to that of concentric in terms of conventional ratios [55]. Second, we were unable to differentiate between central and peripheral causes of fatigue. Future studies should consider incorporating methods to identify the causes of fatigue within this type of protocol. Lastly, the conditions of handball match differ from those created in the simulation protocol used in this study. In handball, many injuries and injury-related situations result from collisions between players, which were not accounted for in the current study. However, this study focused on examining intrinsic factors within individuals identifying intrinsic mechanisms.

## Conclusion

In general, the results of present study demonstrated that a handball match simulation-induced fatigue led to a decrease in the isokinetic peak torque of the hamstring, quadriceps, invertor, and evertor muscles. Furthermore, the findings revealed that fatigue reduced both the H:Q peak torque ratio and the E:I peak torque ratio. These changes could contribute to increased risk of ACL tears and lateral ankle sprains. Considering that hamstring muscles were more affected in the reduction of the H:Q peak torque ratio, it is recommended to focus on strengthening and improving the endurance of this muscle in handball exercises. Additionally, due to the greater degrees of decrease in evertor muscles peak torque associated with the decrease in the E:I peak torque ratio, handball players are advised to prioritize enhancing the endurance and strength of the evertor muscles during their training sessions. Given that most lower limb injuries in handball occur in the final minutes of a match, it is recommended to specifically focus on training of hamstring and evertor muscles to increase their force production and endurance capacity to reduce injury risk and mitigate fatigue-related performance declines.

## Acknowledgments

The authors thank all subjects who volunteered to participate in the present study.

## Author contributions

**Conceptualization:** Ali Shirani Dastjerdi, Hamed Esmaeili, Morteza Sadeghi, Behzad Bashiri.

**Data curation:** Ali Shirani Dastjerdi, Hamed Esmaeili.

**Formal analysis:** Ali Shirani Dastjerdi, Hamed Esmaeili, Morteza Sadeghi, Behzad Bashiri.

**Investigation:** Ali Shirani Dastjerdi, Hamed Esmaeili, Morteza Sadeghi.

**Methodology:** Ali Shirani Dastjerdi, Hamed Esmaeili, Morteza Sadeghi, Behzad Bashiri.

**Project administration:** Ali Shirani Dastjerdi, Hamed Esmaeili, Morteza Sadeghi.

**Supervision:** Hamed Esmaeili.

**Validation:** Hamed Esmaeili.

**Visualization:** Hamed Esmaeili.

**Writing – original draft:** Ali Shirani Dastjerdi, Hamed Esmaeili, Morteza Sadeghi.

**Writing – review & editing:** Hamed Esmaeili, Behzad Bashiri.

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
