## [Decision Letter · Decision Letter 0]

10 Mar 2025

PONE-D-25-07828
Effect of simulated handball match-induced fatigue on isokinetic hamstring-to-quadriceps ratio and invertor-to-evertor ratio in professional players
PLOS ONE

Dear Dr. Esmaeili,

Thank you for submitting your manuscript to PLOS ONE. After careful consideration, we feel that it has merit but does not fully meet PLOS ONE’s publication criteria as it currently stands. Therefore, we invite you to submit a revised version of the manuscript that addresses the points raised during the review process.

We look forward to receiving your revised manuscript.

Kind regards,

Seyed Hamed Mousavi

Academic Editor

PLOS ONE

Reviewers' comments:

Reviewer's Responses to Questions

**Comments to the Author**

1. Is the manuscript technically sound, and do the data support the conclusions?

Reviewer #1: Partly

Reviewer #2: Yes

2. Has the statistical analysis been performed appropriately and rigorously? 

Reviewer #1: Yes

Reviewer #2: Yes

3. Have the authors made all data underlying the findings in their manuscript fully available?

Reviewer #1: Yes

Reviewer #2: Yes

4. Is the manuscript presented in an intelligible fashion and written in standard English?

Reviewer #1: Yes

Reviewer #2: Yes

5. Review Comments to the Author

Reviewer #1: The purpose of this quasi-experimental study was to examine the effects of a functional fatigue protocol in handball on muscle group torque for the knee extensors and flexors and ankle invertors and evertors. There appeared to be a need for this study and the study purpose was clear. More information is needed in the methods section. There is some confusion in the presentation of the results, although figures and tables complement the narrative. More specific details are needed in the discussion and a limitations section is needed as well. A more developed presentation on central and peripheral fatigue is needed in the introduction which will serve a section of the discussion well. There were several word choice and word tense errors, as well as errors and inconsistencies in the reference section. See pdf for details comments and questions.

Reviewer #2: 1) The authors should justify why eccentric torque was not measured or acknowledge it as a limitation.

2) Consider adding physiological fatigue markers (if available) to validate fatigue protocol

3) Discuss whether the reduction in H:Q and E:I ratios reaches clinically significant thresholds

4) if possible, provide effect size interpretations and confidence intervals

5) Ensure consistency in statistical reporting (rounding, decimal places, p-values)

6) if possible, consider adding a sample size power analysis

7) Expand discussion with comparisons to other team sports

8) Provide practical training recommendations for injury prevention

9) You can benefit from the article https://doi.org/10.31459/turkjkin.1576269. in discussion section: When interpreting H:Q ratio reductions and their biomechanical consequences. or in introduction Section: To support the importance of the H:Q ratio in lower limb function.

6. PLOS authors have the option to publish the peer review history of their article (what does this mean?). If published, this will include your full peer review and any attached files.

Reviewer #1: No

Reviewer #2: **Yes: **Esedullah AKARAS

---

## [Author Response · Author response to Decision Letter 1]

27 Mar 2025

Reviewer #1:

The purpose of this quasi-experimental study was to examine the effects of a functional fatigue protocol in handball on muscle group torque for the knee extensors and flexors and ankle invertors and evertors. There appeared to be a need for this study and the study purpose was clear. More information is needed in the methods section. There is some confusion in the presentation of the results, although figures and tables complement the narrative. More specific details are needed in the discussion and a limitations section is needed as well. A more developed presentation on central and peripheral fatigue is needed in the introduction which will serve a section of the discussion well. There were several word choice and word tense errors, as well as errors and inconsistencies in the reference section. See pdf for details comments and questions.

Thank you very much for your thorough evaluation and thoughtful suggestions. We greatly value your feedback, and we believe that the revisions we have made in response to your comments, especially in the methodology section, have significantly strengthened our manuscript. Below, we highlight the key changes implemented in response to your comments:

 Regarding intermittent sport:

Intermittent sport, often referred to as intermittent exercise or interval training, involves alternating periods of high-intensity activity with periods of lower-intensity activity or rest. So, Handball is an intermittent sport.

 Your introduction to the concept of fatigue is superficial and appears to only apply to peripheral fatigue. Please more fully develop the concepts of peripheral and central fatigue first, then perhaps show the readers that the fatigue you are interested in mostly is related to peripheral fatigue. This needs to be well-referenced.

We have now expanded the introduction and discussion sections to include both peripheral and central fatigue.

We have also added the following to the manuscript:

 Sample size calculation

 Inclusion criteria

 General warm up

 A reference on the test-retest reliability of Borg scale measure

 Peak torque per body mass considered for measurements

 A brief description of the problem and the rationale/need for this study before restating the purpose

 Expanded operational definitions of general and local fatigue in the discussion and introduction

 Limitation section

Reviewer #2:

Thank you very much for your detailed review and constructive feedback. We appreciate your insights, which have helped strengthen our manuscript. The comments have been addressed and highlighted in the text. Please see our responses to each comment below.

1) The authors should justify why eccentric torque was not measured or acknowledge it as a limitation.

We have acknowledged this as a limitation in the manuscript.

2) Consider adding physiological fatigue markers (if available) to validate fatigue protocol

We have added physiological markers (Table 1) and expanded the discussion by adding the following:

“The proposed fatigue protocol was designed to induce high levels of physiological and psychological stress, ultimately leading to exhaustion. This was evident from the heart rate (HR) at exhaustion, which reached 186.31 ± 7.4 bpm, approximately 96% of the predicted maximum HR. Additionally, participants reported maximum values on the RPE score of 20, further confirming the extreme cardiovascular stress induced by the protocol. These findings, along with the observed reductions in quadriceps, hamstring invertor and evertor muscles peak torque per body mass and their respective ratios, strongly suggest that the protocol effectively induces fatigue. These results indicate that this protocol reliably simulates fatigue in handball players, making it a valuable tool for both training and research.”

3) Discuss whether the reduction in H:Q and E:I ratios reaches clinically significant thresholds

We have added the following to the discussion section:

“The large effect size (\eta_p^2 > 0.14) demonstrates that the simulated handball game fatigue protocol had a robust and clinically significant impact on knee and ankle musculature peak torques. This large effect size suggests that the simulated handball game fatigue protocol induced significant changes in muscle performance, as measured by peak torques. This is clinically relevant because peak torque is a key indicator of muscle strength and function, which are critical for athletic performance and injury prevention in handball players. For coaches and sports scientists, the large effect size underscores the importance of incorporating fatigue protocols into training regimens to better prepare athletes for the physical demands of competition. By replicating game-like fatigue conditions, athletes can develop greater resilience and adaptability in their knee and ankle musculature.”

4) if possible, provide effect size interpretations and confidence intervals

They have been added to the tables as suggested.

5) Ensure consistency in statistical reporting (rounding, decimal places, p-values)

We have reviewed the statistical reporting for consistency.

6) if possible, consider adding a sample size power analysis

It was added as suggested

7) Expand discussion with comparisons to other team sports

We have included compressions with soccer where it was relevant.

8) Provide practical training recommendations for injury prevention

We have implemented these recommendations in the discussion.

9) You can benefit from the article https://doi.org/10.31459/turkjkin.1576269. in discussion section: When interpreting H:Q ratio reductions and their biomechanical consequences. or in introduction Section: To support the importance of the H:Q ratio in lower limb function.

Thank you so much for providing this valuable reference. We used this reference in the discussion section as suggested:

“Additionally, it has been shown that a decreased H:Q is associated with increased loading on the forefoot and toe regions (50). The increased loading could lead to increased risk of stress fracture in the forefoot region (51).”

---

## [Decision Letter · Decision Letter 1]

10 Apr 2025

PONE-D-25-07828R1
Effect of simulated handball match-induced fatigue on isokinetic hamstring-to-quadriceps ratio and evertor-to-invertor ratio in professional player
PLOS ONE

Dear Dr. Esmaeili,

Thank you for submitting your manuscript to PLOS ONE. After careful consideration, we feel that it has merit but does not fully meet PLOS ONE’s publication criteria as it currently stands. Therefore, we invite you to submit a revised version of the manuscript that addresses the points raised during the review process.

We look forward to receiving your revised manuscript.

Kind regards,

Seyed Hamed Mousavi

Academic Editor

PLOS ONE

Journal Requirements:

Reviewers' comments:

Reviewer's Responses to Questions

**Comments to the Author**

1. If the authors have adequately addressed your comments raised in a previous round of review and you feel that this manuscript is now acceptable for publication, you may indicate that here to bypass the “Comments to the Author” section, enter your conflict of interest statement in the “Confidential to Editor” section, and submit your "Accept" recommendation.

Reviewer #1: All comments have been addressed

Reviewer #2: All comments have been addressed

2. Is the manuscript technically sound, and do the data support the conclusions?

Reviewer #1: Yes

Reviewer #2: Yes

3. Has the statistical analysis been performed appropriately and rigorously? 

Reviewer #1: Yes

Reviewer #2: Yes

4. Have the authors made all data underlying the findings in their manuscript fully available?

Reviewer #1: Yes

Reviewer #2: Yes

5. Is the manuscript presented in an intelligible fashion and written in standard English?

Reviewer #1: No

Reviewer #2: Yes

6. Review Comments to the Author

Reviewer #1: Thank you for addressing the many issues identified. Additional word choice, word tense, and spelling errors remain, some of which were in your highlighted sections so look carefully for my comments. The biggest error was found in the introduction in an incorrect definition of central and peripheral fatigue (see comment and a suggested reference). What was odd was that you correctly discussed central and peripheral fatigue in the discussion section. See pdf for specific comments.

Reviewer #2: (No Response)

7. PLOS authors have the option to publish the peer review history of their article (what does this mean?). If published, this will include your full peer review and any attached files.

Reviewer #1: No

Reviewer #2: **Yes: **Esedullah AKARAS

---

## [Author Response · Author response to Decision Letter 2]

14 Apr 2025

Reviewer #1: Thank you for addressing the many issues identified. Additional word choice, word tense, and spelling errors remain, some of which were in your highlighted sections so look carefully for my comments. The biggest error was found in the introduction in an incorrect definition of central and peripheral fatigue (see comment and a suggested reference). What was odd was that you correctly discussed central and peripheral fatigue in the discussion section. See pdf for specific comments.

Dear reviewer,

Thank you again for reviewing our manuscript and providing your invaluable feedback. We have carefully revised the manuscript to address your suggestions and have clarified the definitions of central and peripheral fatigue in the introduction, incorporating the excellent reference you recommended. Your insights have been incredibly helpful, and we believe these changes have greatly improved the manuscript.

---

## [Editor Report · Decision Letter 2]

16 Apr 2025

Effect of simulated handball match-induced fatigue on isokinetic hamstring-to-quadriceps ratio and evertor-to-invertor ratio in professional player

PONE-D-25-07828R2

Dear Dr. Esmaeili,

We’re pleased to inform you that your manuscript has been judged scientifically suitable for publication and will be formally accepted for publication once it meets all outstanding technical requirements.

Kind regards,

Seyed Hamed Mousavi

Academic Editor

PLOS ONE
---

## [Editor Report · Acceptance letter]

PONE-D-25-07828R2

PLOS ONE

Dear Dr. Esmaeili,

I'm pleased to inform you that your manuscript has been deemed suitable for publication in PLOS ONE. Congratulations! Your manuscript is now being handed over to our production team.

Kind regards,

on behalf of

Dr. Seyed Hamed Mousavi

Academic Editor

PLOS ONE